

# Effects of silver-graphene oxide on seed germination and early growth of crop species

Min-Ji Kim[1], Woong Kim[1] and Haegeun Chung[2]

[1] Department of Materials Science and Engineering, Korea University, Seoul, Republic of Korea
[2] Department of Environmental Engineering, Konkuk University, Seoul, Republic of Korea

## ABSTRACT

Due to its excellent material properties, silver-graphene oxide (Ag-GO) is being studied for diverse applications, such as antimicrobial agents, catalysts and absorbents. Such use of Ag-GO may lead to its release into terrestrial ecosystems, but little is known about the impact of Ag-GO on plants. In the present study, we determined the effects of Ag-GO on seed germination and early growth of crop species by analyzing the germination rate, growth of roots and shoots, hydrogen peroxide ($H_2O_2$) accumulation, and the uptake of Ag in alfalfa, radish and cucumber treated with 0.2–1.6 mg $mL^{-1}$ of Ag-GO. Ag-GO treatment increased the shoot growth of radish at 0.2–1.6 mg $mL^{-1}$ but decreased that of cucumber at 0.8 mg $mL^{-1}$. In addition, Ag-GO enhanced the root elongation of radish at 0.2 mg $mL^{-1}$ but inhibited that of alfalfa at 0.2, 0.8 and 1.6 mg $mL^{-1}$. Ag-GO treatment induced $H_2O_2$ production in alfalfa, radish and cucumber in a concentration-dependent manner. Larger amounts of Ag accumulated in the seedlings as the concentration of Ag-GO increased, and such accumulation suggests that Ag may be transferred to higher trophic levels when plants are exposed to Ag-GO in ecosystems. Our study can, thus, serve as an important basis for setting guidelines for the release of nanomaterials into the environment.

## INTRODUCTION

With the rapid progress of nanotechnology in recent years, nanomaterials are being used in many products, including cosmetics, agrochemicals and biochemical sensors (*Alivisatos, 2004*; *Nair et al., 2010*; *Fleming & Ulijn, 2014*; *Yang et al., 2019*). One of the most widely used nanomaterials for commercial products is silver nanoparticle (Ag NP), owing to their superb material characteristics, including a high antibacterial activity and unique surface plasmon resonance (*Ahamed, AlSalhi & Siddiqui, 2010*; *Vance et al., 2015*). However, Ag NPs can be easily aggregated; therefore, nanocomposites have been formed with carbon materials such as graphene oxide (GO) and photocatalytic materials as supporting materials to maintain the stability of Ag NPs. As the result of the oxidation of graphite, GO with $sp^2$-bonded carbon atoms has many oxygen functional groups, allowing Ag NPs to be formed easily on a sheet of GO. The formation of Ag NPs on a GO

Corresponding author
Haegeun Chung,
hchung@konkuk.ac.kr

sheet not only prevents the agglomeration of Ag NPs, but also enhances their material properties, such as their antibacterial activity and electronic and magnetic properties (*Xu et al., 2011*; *Li et al., 2013*). Due to its excellent properties, silver-graphene oxide (Ag-GO) is increasingly being studied for many applications, for example, as an antibacterial agent for water disinfection, application on membranes for antifouling, catalysts, and electrochemical biosensors (*Bao, Zhang & Qi, 2011*; *Ko et al., 2018*; *Naeem et al., 2019*; *Zhao et al., 2019*). Thus, the potential of Ag-GO for applications in diverse fields warrants investigation into its environmental impacts.

In addition to the abovementioned applications, Ag-GO with high antimicrobial activity has been investigated as a novel antimicrobial agent for the control of crop diseases (*Ocsoy et al., 2013*; *Chen et al., 2016*; *Liang, Yang & Cui, 2017*). Ag-GO has been shown to have a higher antibacterial activity against *Xanthomonas perforans*, a tomato bacterial spot pathogen, than GO or Ag NPs alone. Ag-GO deformed the bacterial structure and destroyed the cell membrane by wrapping the *X. perforans* cell, while exerting no phytotoxicity against tomato leaves (*Ocsoy et al., 2013*). Similarly, 2.5 µg mL$^{-1}$ of Ag-GO had antibacterial activity against bacterial leaf blight of rice caused by *Xanthomonas oryzae* pv. *oryzae (Xoo)*. Ag-GO damaged the structure of the *Xoo* cell wall and membrane, and caused leakage of the cell contents (*Liang, Yang & Cui, 2017*). In addition, Ag-GO deformed the structure of hyphae and induced reactive oxygen species (ROS) in *Fusarium graminearum*, which cause leaf spot disease (*Chen et al., 2016*). Hence, the antimicrobial activity of Ag-GO makes it a potential agent for controlling crop diseases. Given the high potential for the applications of Ag-GO in various fields, it is important to determine the influence of Ag-GO on terrestrial ecosystems, especially on plants that play a central role in ecosystems as a primary producer.

Several studies have demonstrated the effects of Ag NPs on seed germination and plant growth (*Barrena et al., 2009*; *El-Temsah & Joner, 2012*; *Rui et al., 2017*; *Yang et al., 2018*). Treatment with 0.1 mg mL$^{-1}$ of Ag NPs inhibited the root elongation of cucumber by 15%, but it had no significant effect on the root elongation of lettuce (*Barrena et al., 2009*). In another study, treatment with Ag NPs that are 5 nm in diameter reduced the shoot length of flax and barley at 10 mg L$^{-1}$ and that of ryegrass at 10, 20 and 100 mg L$^{-1}$. For Ag NPs that are 20 nm in diameter, the shoot growth of all three plants was inhibited at 20 mg L$^{-1}$; thus, the inhibitory effect of Ag NPs was not size-dependent (*El-Temsah & Joner, 2012*). In addition, inhibited growth was observed in wheat and peanuts grown in soils amended with Ag NPs (*Rui et al., 2017*; *Yang et al., 2018*).

Other reports have shown both positive and negative effects of graphene on plants (*Begum, Ikhtiari & Fugetsu, 2011*; *Zhang et al., 2015*; *Jiao et al., 2016*). Treatment with 2 mg mL$^{-1}$ of graphene for 20 days significantly reduced the growth and biomass of cabbage, red spinach and tomato seedlings. Furthermore, graphene caused oxidative stress and membrane leakage in these species. However, lettuce treated at this dosage of graphene showed no significant difference in plant growth or hydrogen peroxide (H$_2$O$_2$) production (*Begum, Ikhtiari & Fugetsu, 2011*). In contrast, another study demonstrated that graphene at 40 µg mL$^{-1}$ positively affected seed germination and seedling growth of tomato. More specifically, treatment with graphene increased the stem and root lengths

and total fresh weight of tomato seedlings because graphene could penetrate seed husks, resulting in the facilitation of water uptake (*Zhang et al., 2015*). In another study, treatment with 20 mg L$^{-1}$ GO decreased the root length of tobacco compared with that of the control on day 20, but it enhanced the root length of tobacco by two to three times more than that of the control after 35 days. Moreover, GO increased the activity of peroxidase, superoxide dismutase and catalase and decreased the malondialdehyde content in tobacco seedlings (*Jiao et al., 2016*).

Despite the reports on the effects of Ag NPs and GO on various plant species, few studies have been conducted to investigate the impact of Ag-GO on plants. In the present study, we determined the effects of Ag-GO on seed germination and early growth of three plant species, alfalfa, radish and cucumber. Alfalfa, used as fodder, plays an important role in agricultural ecosystems by fixing nitrogen. Radish and cucumber are vegetable crops that have been previously used to investigate the effects of nanomaterials (*Wierzbicka & Obidzińska, 1998*; *Migliore, Cozzolino & Fiori, 2003*). We examined changes in the germination rate and shoot and root lengths to investigate the potential effects of Ag-GO on germination and seedling growth and measured the H$_2$O$_2$ content to confirm the oxidative stress due to ROS. In addition, the Ag content in the seedlings of the three species was measured to determine the uptake of Ag upon Ag-GO treatment.

## MATERIALS AND METHODS

### Synthesis and characterization of Ag-GO

Graphene oxide was prepared using a modification of Hummers' method described elsewhere (*Hummers & Offeman, 1958*; *Chung et al., 2015*) and Ag-GO was synthesized by the glucose reduction method (*Tang et al., 2013*; *Kim et al., 2018*). Briefly, 50 mL of an aqueous GO solution (0.5 mg mL$^{-1}$) was ultrasonicated for 1 h using a bar sonicator (VC 750 ultrasonic processor; Sonic & Materials Inc., Newtown, CT, USA). The GO solution was mixed uniformly with 10 mL polyvinylpyrrolidone (PVP) solution (4 mg mL$^{-1}$, average molecular weight ~29,000, Sigma–Aldrich, St. Louis, MO, USA) and 800 mg glucose (Sigma–Aldrich, St. Louis, MO, USA) and the solution was heated at 45 °C. Subsequently, silver-ammonium hydroxide solution (10 mL) was added and the solution was held at 45 °C for 7 min. The mixture was centrifuged at 12,298×$g$ for 10 min and the supernatant was decanted; the remaining pellet was washed several times with deionized (DI) water to remove excess ions and dried at 80 °C.

The microscopic morphology of Ag-GO was observed by transmission electron microscopy (Tecnai 20; FEI, Hillsboro, ON, USA). Absorption spectra were recorded using a UV–VIS spectrophotometer (Varian Cary 50; Cary, NC, USA). Raman spectroscopy was conducted using a LabRAM ARAMIS IR2 (Horiba, Kyoto, Japan) to analyze the defects and strain in Ag-GO and GO. X-ray photoelectron spectroscopy (XPS) spectra were measured with an automated XPS Microprobe (PHI 5000 Versa Probe; ULVAC-PHI, Chigasaki, Kanagawa, Japan) to compare the surface bonding before and after deposition of Ag NPs on the GO sheet; the ratio of Ag to GO was calculated through XPS.

## Seed exposure

Three plant species, alfalfa (*Medicago sativa* L.), radish (*Raphanus sativus* L.) and cucumber (*Cucumis sativus* L.), were used in our study, and the seeds of these plants which were stored after harvest were purchased from Danong Inc., Namyangju-si, South Korea. Before exposure to the nanomaterials, seeds were soaked in a 10% sodium hypochlorite solution for 10 min to sterilize them and rinsed with DI water three times (*US Environmental Protection Agency, 2012*). Ten sterilized seeds were transferred onto filter paper placed in a $100 \times 15$ mm Petri dish containing 5 mL of Ag-GO suspension at different concentrations (0, 0.2, 0.4, 0.8 and 1.6 mg mL$^{-1}$). The different concentrations were prepared by diluting the stock Ag-GO suspension of 1.6 mg mL$^{-1}$. Additionally, 1.6 mg mL$^{-1}$ of GO and Ag NPs were applied to seeds to compare their effects to those of Ag-GO. Ag NPs (Sigma–Aldrich, St. Louis, MO, USA) were stabilized using PVP and their diameter was $70.2 \pm 4.9$ nm (*Yasur & Rani, 2013*). Petri dishes were sealed with parafilm and incubated at 25 °C in the dark for 7 days. Four replicates of each treatment were analyzed, for a total of 40 seeds per test species. During incubation, the germination percentage was determined every 2 days by counting the number of germinated seeds that had roots longer than 2 mm; the shoot and root lengths were measured at the end of the incubation period using a ruler.

## Analysis of $H_2O_2$ in seedlings

The Amplex® red hydrogen peroxide/peroxidase assay kit (Invitrogen™, Carlsbad, CA, USA) was used for the determination of $H_2O_2$ on day 7 at the end of the experiment. Seedlings were washed with DI water to remove nanomaterials on the surface of the seedlings. Adhering moisture was removed from the seedlings before grinding them in liquid nitrogen using a mortar and pestle. A total of 50 mg of tissue powder were resuspended in 250 µl of cold phosphate buffer (20 mM $K_2HPO_4$, pH 6.5). After centrifugation at 4 °C, 50 µl of supernatant was incubated with 0.2 U mL$^{-1}$ horseradish peroxidase and 100 µm Amplex red reagent (10-acetyl-3,7-dihydrophenoxazine) at room temperature in the dark for 30 min. Subsequently, the fluorescence was measured with a Synergy HT multi-mode microplate reader (Biotek, Winooski, VT, USA) (excitation at 530 nm and emission at 590 nm).

## Quantification of silver content in fresh seedlings

Ag content in seedlings was quantified on day 7, at the end of the experiment, following the method described by *Geisler-Lee et al. (2013)* with modifications. Seedlings of the three species were washed with DI water and their fresh weight (F.W.) was measured. Seedlings were fully digested using a mixture of 2 mL of $HNO_3$ used for trace-metal analysis (70%, Sigma–Aldrich, St. Louis, MO, USA) and 1 mL Co standard solution (1,000 mg mL$^{-1}$ in nitric acid, Sigma–Aldrich, St. Louis, MO, USA) heated at 120 °C for 30 min. After the solution cooled down, 2 mL of $H_2O_2$ used for trace analysis ($H_2O_2$, ~30% w/w, Sigma–Aldrich, St. Louis, MO, USA) was added and the mixture was heated at 120 °C for 30 min. The digested samples were diluted with DI water to 20 mL and filtered using a 0.22-µm polyvinylidene fluoride (PVDF) syringe filter. Each diluted sample

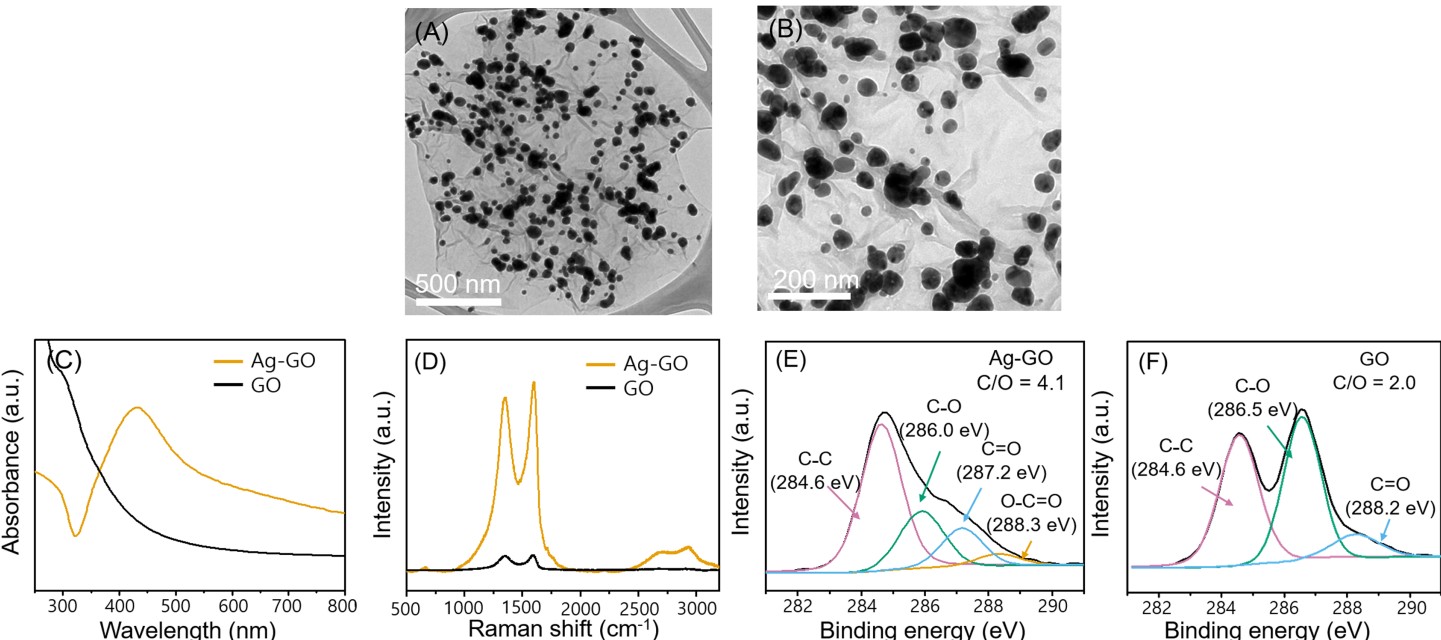

**Figure 1 Characterization of Ag-GO and GO.** (A) and (B) Transmission electron microscope (TEM) images of Ag-GO. (C) Ultraviolet–visible (UV–VIS) spectrum, (D) Raman spectrum, X-ray photoelectron spectroscopy (XPS) C1s spectra and binding energy of (E) GO and (F) Ag-GO.

was analyzed using an inductively coupled plasma-optical emission spectrometer (ICP-OES, iCAP6300 Duo; Thermo Scientific, Waltham, MA, USA).

## Statistical analyses

Statistical analyses were carried out using SAS® version 9.4 (SAS Inc., Cary, NC, USA) to test the effects of Ag-GO, Ag NPs and GO on seed germination. One-way analyses of variance (ANOVA) were employed to determine the effects of Ag-GO treatment on germination percentage, the lengths of the root and shoot, $H_2O_2$ content, and Ag content in seedlings. Additionally, one-way ANOVA was conducted to compare the effects of treatments with Ag-GO, Ag NPs and GO at a concentration of 1.6 mg mL$^{-1}$. The effects of the treatments were accepted as significant at $\alpha = 0.05$. Tukey's honestly significant difference test was conducted to further investigate which means differed from other means within a group ($p < 0.05$) when the effect of each treatment was significant for the parameters studied. In addition, regression analyses were performed to determine the relationship between the concentration of Ag-GO treatment and Ag content in seedlings treated with Ag-GO.

## RESULTS

### Characterization of Ag-GO

The TEM images of Ag-GO showed that the Ag NPs were formed on the wrinkled surface of GO (Figs. 1A and 1B). The average diameter of Ag NPs was 57.5 ± 1.9 (average ± one standard error; $n = 250$) nm. The UV–VIS spectrum of GO (Fig. 1C) displayed no

**Table 1 Germination percentage (%) of seedlings treated with Ag-GO, Ag NPs, and GO (average ± one standard error, $n = 4$).**

| | Alfalfa | | | Radish | | | Cucumber | | |
|---|---|---|---|---|---|---|---|---|---|
| | Day 2 | Day 4 | Day 7 | Day 2 | Day 4 | Day 7 | Day 2 | Day 4 | Day 7 |
| DI water only | 100 | 100 | 100 | 80 ± 8 | 93 ± 5 | 100 | 100 | 100 | 100 |
| 0.2 mg mL$^{-1}$ of Ag-GO | 87 ± 6 | 100 | 100 | 89 ± 4 | 97 ± 3 | 100 | 100 | 100 | 100 |
| 0.4 mg mL$^{-1}$ of Ag-GO | 72 ± 12 | 100 | 100 | 91 ± 3 | 95 ± 4 | 95 ± 4 | 96 ± 2 | 100 | 100 |
| 0.8 mg mL$^{-1}$ of Ag-GO | 87 ± 4 | 100 | 100 | 73 ± 6 | 90 ± 5 | 100 | 100 | 100 | 100 |
| 1.6 mg mL$^{-1}$ of Ag-GO | 90 ± 4 | 100 | 100 | 87 ± 6 | 100 | 100 | 100 | 100 | 100 |
| 1.6 mg mL$^{-1}$ of Ag NPs | 90 ± 4 | 100 | 100 | 87 ± 5 | 97 ± 3 | 100 | 100 | 100 | 100 |
| 1.6 mg mL$^{-1}$ of GO | 100 | 100 | 100 | 100 | 100 | 100 | 100 | 100 | 100 |

obvious absorption band in the wavelength range of 300–800 nm. In contrast, the UV–VIS spectrum of Ag-GO showed maximum absorbance at 430 nm, indicating that Ag NPs were loaded on the GO sheets. The Raman spectra of Ag-GO and GO (Fig. 1D) showed two peaks, the D band at 1,350 cm$^{-1}$ and the G band at 1,590 cm$^{-1}$. However, the intensity of the D and G bands in the Raman spectrum of Ag-GO was enhanced by surface-enhanced Raman scattering. The $I_D/I_G$ ratios of GO and Ag-GO were 0.95 and 0.92, respectively. Analysis of surface bonding using XPS (Figs. 1E and 1F) clearly showed that the intensity of oxygenated carbon bonds decreased after Ag NP formation on GO. The XPS C1s spectra of GO showed three main peaks, for C–C, C–O and C=O bonds. The C–O and C=O bond peaks of Ag-GO showed a relatively lower intensity than those of GO. Additionally, the carbon-to-oxygen ratio increased from 2.0 to 4.1 after Ag NP formation on the GO sheet. The reduction of oxygenated carbon after Ag NP formation on a GO sheet has also been demonstrated by other groups (*Tang et al., 2013*; *Kim et al., 2018*). The weight ratio of Ag to GO was approximately 0.64.

## Effects of Ag-GO on seed germination and seedling growth

Treatment with Ag-GO had no significant effect on germination percentage (Table 1). In addition, germination percentage was not altered by Ag NP or GO treatment. Regardless of the Ag-GO treatment concentration, 100% of the alfalfa and cucumber seeds germinated on day 4 and day 7, respectively, and 95–100% of radish seeds germinated on day 7. Ag-GO had both positive and negative effects on seedling growth (Fig. 2). The shoot length of alfalfa was not affected significantly by Ag-GO treatment; however, that of radish and cucumber showed a significant response to treatment with Ag-GO (Fig. 2A). More specifically, the shoot length of radish treated with Ag-GO at 0.2 to 1.6 mg mL$^{-1}$ increased up to 52% ($p < 0.0001$) and that of radish treated with Ag NPs at 1.6 mg mL$^{-1}$ increased by 36% when compared to the control. However, the shoot length of cucumber decreased by 16% upon treatment with Ag-GO at 0.8 mg mL$^{-1}$ compared to the control ($p = 0.0003$).

Ag-GO treatment also had a significant effect on the root growth of alfalfa and radish (Fig. 2B). Treatment with 1.6 mg mL$^{-1}$ of Ag-GO inhibited the root elongation of alfalfa by 41% compared to that of the control ($p = 0.0011$). However, the root length of radish

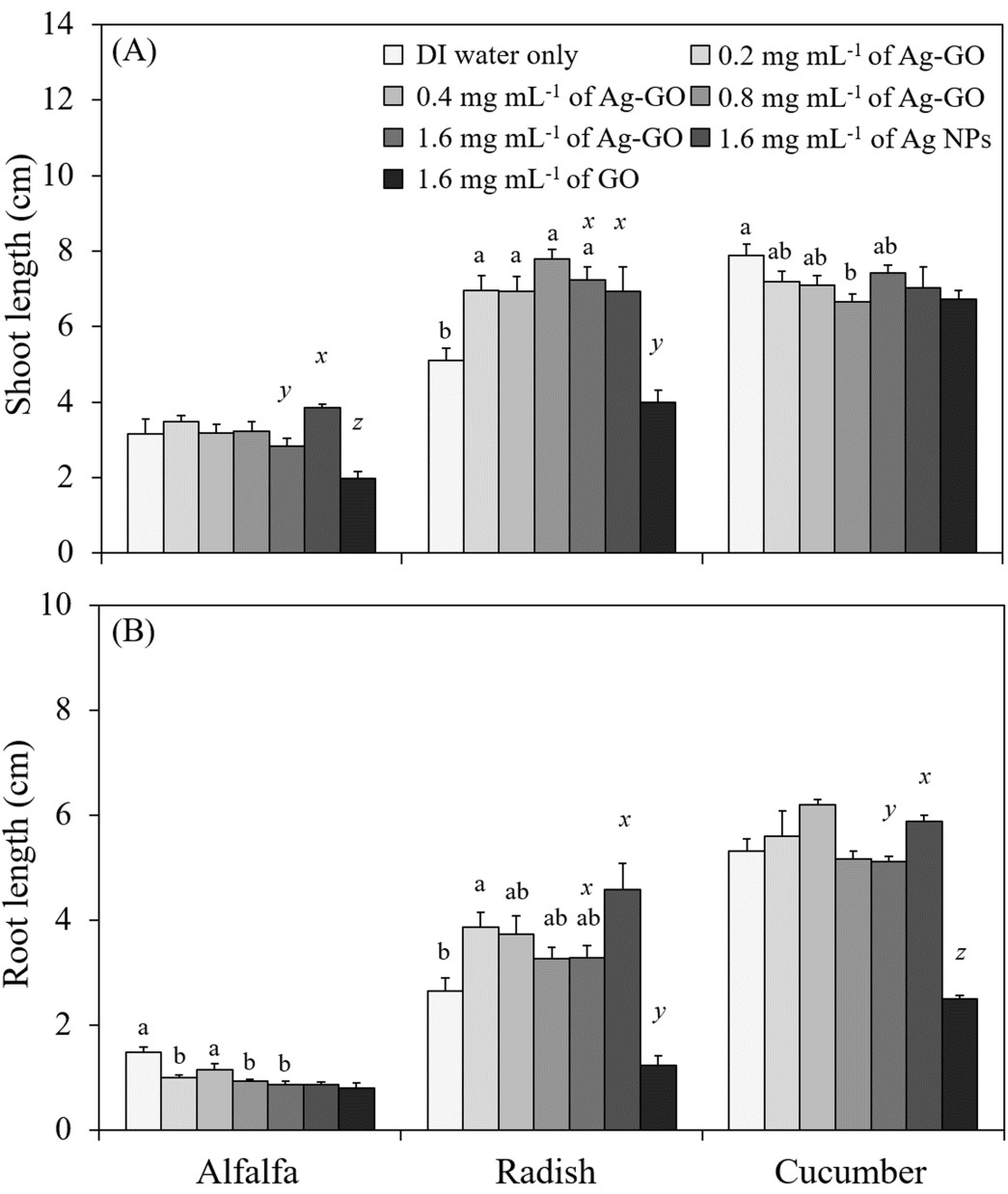

**Figure 2 Effects of Ag-GO on the growth of alfalfa, radish and cucumber seedlings.** (A) Shoot length and (B) root length of seedlings at day 7. Means followed by the same letter are not significantly different at $\alpha = 0.05$. Letters a and b indicate a significant difference among Ag-GO treatments at different concentrations, and $x$, $y$ and $z$ indicate that among 1.6 mg mL$^{-1}$ of Ag-GO, Ag NP and GO treatments. Error bars represent one standard error ($n = 4$).

treated with 0.2 mg mL$^{-1}$ of Ag-GO increased by 45% compared to that of the control ($p = 0.0456$) and its effect on root elongation significantly differed from that of 1.6 mg mL$^{-1}$ of GO ($p = 0.0002$). In contrast, treatment with Ag-GO had no significant effect on the root length of cucumber. Overall, the effects of Ag-GO, Ag NPs and GO treatments on root elongation differed significantly ($p < 0.001$).
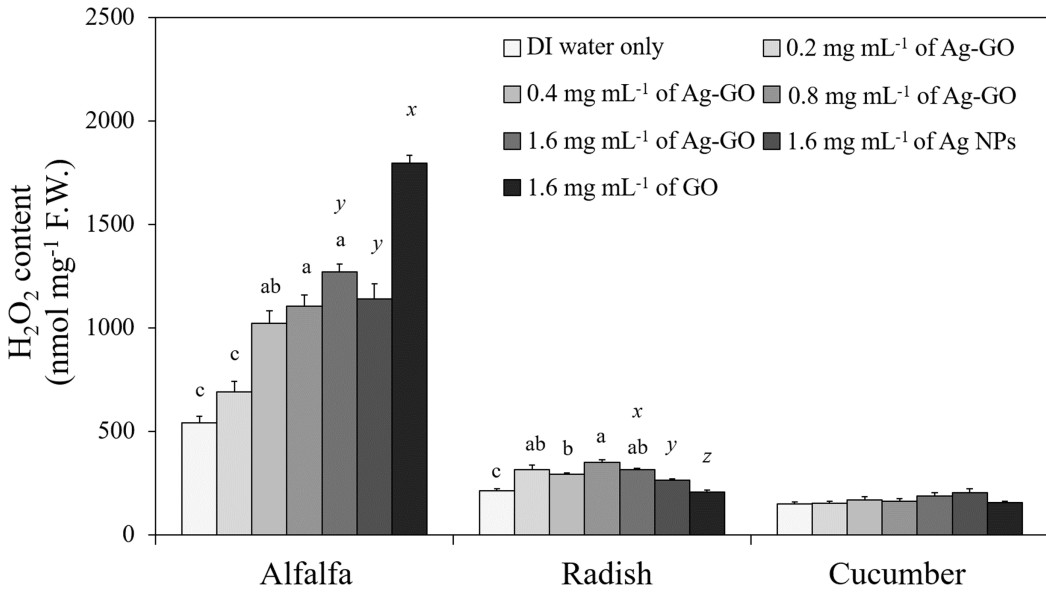

**Figure 3 Effects of Ag-GO on the accumulation of H$_2$O$_2$ in alfalfa, radish and cucumber seedlings at day 7.** Means followed by the same letter are not significantly different at $\alpha$ = 0.05. Letters a, b and c indicate a significant difference among Ag-GO treatments at different concentrations, and *x*, *y* and *z* indicate that among 1.6 mg mL$^{-1}$ of Ag-GO, Ag NP and GO treatments. Error bars represent one standard error ($n$ = 4).

## H$_2$O$_2$ content in seedlings treated with Ag-GO

Ag-GO treatment significantly increased the H$_2$O$_2$ content in alfalfa and radish seedlings (Fig. 3). When alfalfa was treated with 1.6 mg mL$^{-1}$ of Ag-GO, its H$_2$O$_2$ content was more than twice that of the control ($p < 0.0001$) and similar to that in alfalfa treated with 1.6 mg mL$^{-1}$ of Ag NPs ($p < 0.0001$). In radish, the production of H$_2$O$_2$ was increased by treatment with Ag-GO compared to that in the control ($p < 0.0001$). In addition, the level of H$_2$O$_2$ significantly differed for Ag-GO, Ag NPs and GO treatments ($p < 0.0001$). In contrast, the content of H$_2$O$_2$ in cucumber seedlings was not altered by Ag-GO, Ag NPs, or GO treatments.

## Ag content of seedlings treated with Ag-GO

The results of the regression analysis showed that the concentration of Ag-GO and the Ag content in the three seedlings were positively correlated. For example, the Ag content in alfalfa was positively correlated with the treatment concentration of Ag-GO ($p < 0.001$, $R^2$ = 0.85). Likewise, in the case of radish and cucumber, there was positive correlation between the concentration of Ag-GO and Ag content in these species ($p = 0.004$, $R^2$ = 0.71 in radish and $p < 0.0001$, $R^2$ = 0.94 in cucumber). According to the ANOVA results, there were no significant differences in the Ag content in alfalfa and radish at the concentrations of 0.2, 0.4 and 0.8 mg mL$^{-1}$ of Ag-GO. However, at 1.6 mg mL$^{-1}$ of Ag-GO, Ag accumulation in alfalfa and radish was significantly higher than that for the other treatments ($p < 0.0001$ and $p = 0.0059$, respectively). In cucumber, there was no significant difference in Ag content between 0.2 and 0.4 mg mL$^{-1}$ of Ag-GO treatments, but the Ag content was significantly higher at 0.8 and 1.6 mg mL$^{-1}$ of Ag-GO ($p < 0.001$)

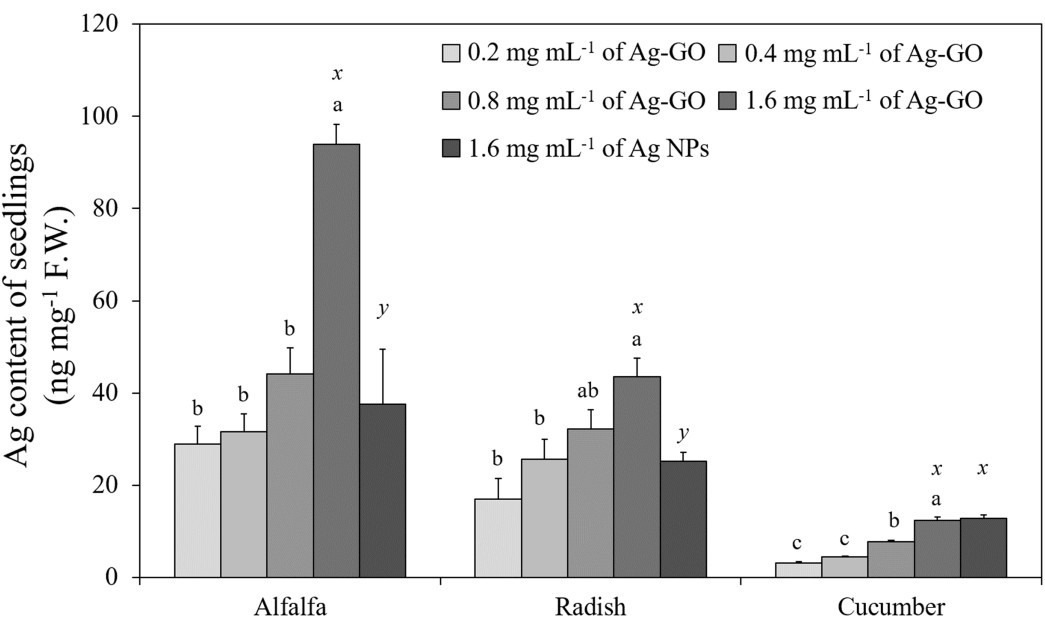

**Figure 4 Concentration of Ag in seedlings exposed to Ag-GO at 0.2, 0.4, 0.8 and 1.6 mg mL$^{-1}$ and Ag NPs at 1.6 mg mL$^{-1}$ for 7 days (average ± one standard error, $n = 4$).** The letters a, b and c denote a significant difference among Ag-GO treatments and x and y denote that between 1.6 mg mL$^{-1}$ of Ag-GO and 1.6 mg mL$^{-1}$ of Ag NP treatments.

(Fig. 4). The Ag contents in alfalfa and radish treated with 1.6 mg mL$^{-1}$ of Ag-GO were 93.9 ± 11.6 ng mg$^{-1}$ F.W. and 43.6 ± 2.0 ng mg$^{-1}$ F.W., respectively. The Ag content in cucumber treated with 1.6 mg mL$^{-1}$ of Ag-GO was 12.5 ± 1.5 ng mg$^{-1}$ F.W., which was four times the content of Ag in cucumber treated with 0.2 mg mL$^{-1}$ of Ag-GO. Furthermore, the Ag content for Ag-GO and Ag NP treatments was significantly different in alfalfa and radish. The Ag content in alfalfa and radish exposed to 1.6 mg mL$^{-1}$ of Ag-GO was approximately two times that of seedlings treated with 1.6 mg mL$^{-1}$ of Ag NPs ($p < 0.0049$ and $p = 0.0079$, respectively). However, the Ag content in cucumber was similar for both treatments (Fig. 4).

## DISCUSSION

The effects of Ag-GO on the early growth of plants were dependent on the plant species with different seed size. The three species used in the present study are from different plant families and have seeds of different sizes, and they showed distinct responses to Ag-GO. Root elongation of the small-seeded alfalfa and radish were enhanced and inhibited, respectively, by Ag-GO treatment, but that of the large-seeded cucumber was not affected. Similarly, in studies that have determined the effects of multi-walled carbon nanotubes, the germination of small-seeded species such as lettuce were affected whereas large-seeded species such as cucumber were not affected by multi-walled carbon nanotube treatment. Therefore, the difference in seed size might be a factor that accounts for the difference in the response of plants to nanomaterials (*Cañas et al., 2008*; *Begum et al., 2012*). Additionally, according to previous studies, the effects of nanomaterials on plants depend on the characteristics of the nanomaterials and the xylem anatomy of the plants, which

affect the absorption of the nanomaterials onto the root (*Cañas et al., 2008*; *Begum, Ikhtiari & Fugetsu, 2011*). *Begum, Ikhtiari & Fugetsu (2011)* showed that the response of cabbage, tomato, red spinach and lettuce to graphene depends on the plant species, because of differences in root systems. High concentrations of graphene inhibited the growth of cabbage, tomato and red spinach, but had no significant effect on lettuce. These results are likely due to the difference in the uptake of nanomaterials by distinctive xylem structures. Likewise, *Cañas et al. (2008)* reported that the response of plants to nanomaterials varies among plant species and that it depends on the characteristics of nanomaterials.

In our study, treatment with Ag-GO and Ag NPs increased the length of shoots and roots in radish but had no effect on root length in cucumber. However, in another study, radish and cucumber treated with Ag NPs showed different responses compared to our results, owing to the properties of the nanomaterials used. More specifically, Ag NPs with a size in the range of 1–10 nm decreased elongation of the shoot and root when applied at a concentration of 0.5 mg mL$^{-1}$ (*Zuverza-Mena et al., 2016*). Altogether, the responses of the plants vary depending on the characteristics of the nanomaterials, such as the particle size.

Treatment with Ag-GO led to the production of $H_2O_2$ during early seedling growth, but relatively low levels of $H_2O_2$ may enhance seedling growth. Plants produce several kinds of ROS as byproducts of aerobic metabolic pathways, and excessive ROS can cause oxidative damage to proteins, DNA, lipids and finally cell death (*Apel & Hirt, 2004*). Several studies demonstrated that nanomaterials produced $H_2O_2$ and induced oxidative stress in plants (*Begum, Ikhtiari & Fugetsu, 2011*; *Anjum et al., 2014*). *Begum, Ikhtiari & Fugetsu (2011)* showed that graphene induced dose-dependent $H_2O_2$ accumulation in cabbage, tomato and red spinach at concentrations of 500, 1,000 and 2,000 mg L$^{-1}$ using the ROS-sensitive probe 2′,7′-dichlorofluorescein diacetate (DCFH-DA). It induced oxidative stress by causing cell membrane damage and electrolyte leakage in leaves. According to our results, high concentrations of $H_2O_2$ in alfalfa induced by Ag-GO likely inhibited root growth. However, the relatively low levels of $H_2O_2$ in radish and cucumber weakly affected the seedling growth. In a previous study, *Anjum et al. (2014)* demonstrated that the activity of catalase and ascorbate peroxidase in faba bean (*Vicia faba* L.) seedlings was increased by treatment with 400 and 800 mg L$^{-1}$ of graphene, and this prevents the overproduction of $H_2O_2$ in faba bean seedlings. Consequently, the low level of $H_2O_2$ accumulated in faba bean seedlings could increase the absorption of liquid by seeds, enhancing the emergence and length of the root.

Seedlings treated with Ag-GO accumulated more Ag at higher concentrations of Ag-GO. Previous studies have also demonstrated that Ag uptake during seedling growth is dose-dependent, but the amount of Ag uptake is different among plant species, treatment concentrations and the physicochemical properties of Ag NPs (*Geisler-Lee et al., 2013*; *Zuverza-Mena et al., 2016*). *Geisler-Lee et al. (2013)* demonstrated that in *Arabidopsis thaliana*, Ag NPs having a size of 40 nm translocated from border cells to internal root tissue and vascular tissue and that Ag accumulation was dose-dependent. Likewise, the Ag content in radish seedlings treated with 2 nm Ag NPs was concentration-dependent

(*Zuverza-Mena et al., 2016*). This tendency was also observed in tomatoes and castor seeds treated with Ag NPs, but the accumulated amount differed according to several factors, such as the culture medium, seed size or coat and physicochemical properties of the nanomaterials (*Song et al., 2013*; *Yasur & Rani, 2013*). Interestingly, the difference between Ag accumulation in seedlings treated with Ag-GO and those treated with Ag NPs was distinct for small-seeded species. Further study is needed to determine whether Ag accumulation is due to the Ag NPs on the GO sheet or due to $Ag^+$ and to investigate the interaction between Ag-GO and the seedling surface.

## CONCLUSIONS

We demonstrate here, that Ag-GO can significantly affect the early growth of plants in a species-specific manner. The shoot growth of cucumber and the root growth of alfalfa was inhibited by Ag-GO treatment, whereas the growth of roots and shoots in radish was enhanced by exposure to Ag-GO. The uptake of Ag from Ag-GO by seedlings was dose-dependent. The production of $H_2O_2$ increased with the concentration of Ag-GO, and high levels of $H_2O_2$ may account for the inhibited growth of alfalfa seedlings. Further study to investigate other ROS and series of antioxidative mechanisms activated by Ag-GO and long-term experiments employing various plant species are necessary before Ag-GO can be utilized more widely in diverse fields.

### Funding
This work was supported by the Basic Science Research Program through the National Research Foundation of Korea (NRF), which is funded by the Ministry of Education (2016R1D1A1B03931560). The funders had no role in study design, data collection and analysis, decision to publish, or preparation of the manuscript.

### Grant Disclosures
The following grant information was disclosed by the authors:
Ministry of Education: 2016R1D1A1B03931560.

### Competing Interests
The authors declare that they have no competing interests.

### Author Contributions
- Min-Ji Kim conceived and designed the experiments, performed the experiments, analyzed the data, prepared figures and/or tables, authored or reviewed drafts of the paper, and approved the final draft.
- Woong Kim analyzed the data, authored or reviewed drafts of the paper, and approved the final draft.
- Haegeun Chung conceived and designed the experiments, analyzed the data, authored or reviewed drafts of the paper, and approved the final draft.

## Data Availability

The raw measurements are available in the Supplemental Files.

## Supplemental Information

Supplemental information for this article can be found online at http://dx.doi.org/10.7717/peerj.8387#supplemental-information.

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
