# Peer review of "Effects of silver-graphene oxide on seed germination and early growth of crop species"

_PeerJ, doi:10.7717/peerj.8387_

## Round 0.1 · original submission · Major Revisions

Although one of the reviewers has recommended that your manuscript be rejected, I believe you can satisfy this reviewers concerns by providing a much stronger justification for the concentration of nanomaterials tested. Please address this and the other reviewer comments. When you have made all of the revisions to your manuscript, I strongly suggest that you have the manuscript edited by a native English speaker to improve its clarity.

Reviewer 1 ·

Basic reporting

This study is very clear and the professional English is used throughout. But literature references are not sufficient about Ag NPs phytotoxicological:
Yang Jie, Jiang Fuping, Ma Chuanxin, et al., 2018. Alteration of Crop Yield and Quality of Wheat upon Exposure to Silver Nanoparticles in a Life Cycle Study. JOURNAL OF AGRICULTURAL AND FOOD CHEMISTRY, 66(11): 2589-2597.
Rui Mengmeng, Ma Chuanxin, Tang Xinlian, et al., 2017. Phytotoxicity of Silver Nanoparticles to Peanut (Arachis hypogaea L.): Physiological Responses and Food Safety. ACS SUSTAINABLE CHEMISTRY & ENGINEERING, 5(8): 6557-6567.

Experimental design

Research question well defined, relevant and meaningful. Methods described with sufficient detail and information to replicate.

Validity of the findings

This study is very novel and interesting, especially for three different plants.

Additional comments

It is a good and novle research. I think I don't think it's necessary to set up so many concentrations, especially 1.6 mg/ml, which is too high.

·

Basic reporting

The authors have made an attempt to study the effect of Ag GO on the growth of three different plant species in right earnest. The observed results appear to be convincing and original.

Experimental design

The number of seeds taken for germinability testing is small (10 seeds per plate x 4). Generally for this type of study a larger number seeds, at least 50 seeds per plate, would have been a better proposition to study the germination.

The authors have not mentioned if they have used freshly harvested seeds or the ones stored after harvesting.

Validity of the findings

There are several reports on the effect of Ag NPs on seed germination and growth of plants however the effect of Ag+GO on the growth parameters of alfalfa, radish and cucumber has been a novel attempt. The results obtained in this study are valid and new to the understanding of plant growth.

Additional comments

In the present study it has been found that alfalfa accumulates relatively higher quantity of Ag when compared to other two species. The chosen plant species such as alfalfa, radish and cucumber are eaten as vegetables.Wouldn't be toxic to human cells? Have the authors made any effort to evaluate the toxicity of Ag, GO and Ag+GO?
The authors have concluded that the size of the seeds plays role in the growth which is affected by Ag+GO. Are there any evidence to prove this fact?
Have the authors determined the accumulation of Ag GO just after imbibition? may be after 5-6 hours?

Reviewer 3 ·

Basic reporting

I appreciate the authors for the attempt to elucidate the Ag-GO induced phytotoxicity in differed xylem structured plant species.

Experimental design

Nanotoxicity assessment of the plants is a significant and serious environmental and agriculture concern over the present epoch of the nano revolution. With that similar memorandum, the toxicity studies to the plants should be thoroughly executed with well-designed experiments. However, most of the phytotoxicity studies to nanomaterials determined by the higher concentration of nanomaterials treated. And I feel this manuscript is a similar one. It is not acceptable limits to the testing of phytotoxicity because there is a lot of experimental evidence already proved that induced phytotoxicity by higher concentrations of nanomaterials which is the real characteristic feature of nanomaterials and it must do toxicity to various organisms treated.

Validity of the findings

My expectation towards the manuscript was failed because this manuscript somewhat failed to prove the scientific innovations and merits and found no valid analytical with quantitative experiments executed regarding the phytotoxicity of treated seedlings.

Additional comments

This research study performed like another regular study, hence I believed, this manuscript is not very suitable for PeerJ readers.

---

## Round 0.2 · Minor Revisions

Please consider whether incorporation of the references indicated by Reviewer 3 would be appropriate. Also, you may wish to submit documentation that the manuscript was professionally edited.

Reviewer 1 ·

Basic reporting

GOOD

Experimental design

GOOD

Validity of the findings

GOOD

Additional comments

Authors have addressed all my comments.

Reviewer 3 ·

Basic reporting

The manuscript does not fulfill my expectations to address the basic and quantitative reporting of using nanomaterials in the biological content. Moreover, authors justified the reviewer comments not in the accepted level.

Experimental design

It doesn't meet the standard analytical methods as much as concern about biological content. I suggest to refer some of the literature which authors can find some useful information in the analytical tool that can be applied in their research data.

1) http://www.sciencedirect.com/science/article/pii/S1878535216301666

2) https://doi.org/10.3390/agronomy9100610

Validity of the findings

The novelty of this findings is less conclusive.

---

## Round 0.3 · accepted · Accept

Thank you for your hard work in revising your manuscript.